# Chiral Ag23 nanocluster with open shell electronic structure and helical face-centered cubic framework

Chao Liu[1], Tao Li[2], Hadi Abroshan [3], Zhimin Li[1], Chen Zhang[4], Hyung J. Kim [3,5], Gao Li[1] & Rongchao Jin[3]

We report the synthesis and crystal structure of a nanocluster composed of 23 silver atoms capped by 8 phosphine and 18 phenylethanethiolate ligands. X-ray crystallographic analysis reveals that the kernel of the Ag nanocluster adopts a helical face-centered cubic structure with $C_2$ symmetry. The thiolate ligands show two binding patterns with the surface Ag atoms: tri- and tetra-podal types. The tetra-coordination mode of thiolate has not been found in previous Ag nanoclusters. No counter ion (e.g., $Na^+$ and $NO_3^-$) is found in the single-crystal and the absence of such ions is also confirmed by X-ray photoelectron spectroscopy analysis, indicating electrical neutrality of the nanocluster. Interestingly, the nanocluster has an open shell electronic structure (i.e., $23(Ag\ 5s^1)$–$18(SR) = 5e$), as confirmed by electron paramagnetic resonance spectroscopy. Time-dependent density functional theory calculations are performed to correlate the structure and optical absorption/emission spectra of the Ag nanocluster.

[1] State Key Laboratory of Catalysis, Dalian Institute of Chemical Physics, Chinese Academy of Sciences, Dalian 116023, P. R. China. [2] School of Physical Science and Technology, ShanghaiTech University, Shanghai 201210, P. R. China. [3] Department of Chemistry, Carnegie Mellon University, Pittsburgh, PA 15213, USA. [4] Department of Chemistry, University of Missouri-Columbia, Columbia, MO 65211, USA. [5] School of Computational Sciences, Korea Institute for Advanced Study, Seoul 02455, Korea. Correspondence and requests for materials should be addressed to G.L. (email: gaoli@dicp.ac.cn) or to R.J. (email: rongchao@andrew.cmu.edu)

Silver nanoparticles (AgNPs) constitute an important type of nanomaterial for a variety of innovative applications[1–4]. As compared with the bulk counterpart, the nanoparticles possess higher surface-to-volume ratios and chemical potential. These characteristics lead to unique optical, electrical, and thermal properties, which constitute the basis of novel applications in sensing, catalysis, nanoelectronics, bio-tagging, etc[5–7]. The anti-bactericidal and antifungicidal activities of AgNPs have gained an increasing interest for their applications in coatings, textiles, wound treatment, sterilization, and biomedical devices[8, 9]. The use of nano-silver reduces cytotoxicity but not antibacterial/antifungal efficacy—an attractive feature attributed to the formation of free radicals from the surface of the nanoparticles[10].

The properties of AgNPs are strongly influenced by their morphology, size, shape, aggregation state, and surface engineering[2, 11]. Therefore, systematic development of AgNPs is required for optimization of the nanoparticles for specific applications, e.g., selective destruction of cancer cells[12]. In this regard, significant efforts have been made to synthesize and characterize ultrasmall-sized Ag metalloid and Ag(I) nanoclusters[13–24]. Studies have revealed important size-dependent features. For example, some nanoclusters with specific sizes have excellent stability and photoluminescence properties[19–21]. Despite the great advances in the synthesis, the total structure determination of Ag nanoclusters still remains to be a major challenge. Thus far, only a few structures of Ag nanoclusters have been resolved by X-ray crystallography[16–24].

Ultrasmall, atomically precise nanoclusters are of great importance in fundamental research for deep understanding of the chemical nature of nanomaterials[1, 11]. Well-defined silver nanoclusters with fully resolved crystal structures offer unique platforms for accurate electronic structure calculations to shed light on the structure–activity relationship. Such nanoclusters also offer insights into the origin of differences in nano- and macroscale metal materials. For example, the plasmonic Ag nanoparticles are well known to adopt a face-centered cubic (fcc) structure[2]. Recently, nanoclusters of $Ag_{14}(SC_6H_3F_2)_{12}(PPh_3)_8$, $[Ag_{62}S_{12}(S^t\text{-}Bu)_{32}]^{2+}$, and $[Ag_{67}(SPhMe_2)_{32}(PPh_3)_8]^{3+}$ with fcc-type structures have also been reported[16, 19, 20], and Yang et al. reported the large structures of plasmonic $Ag_{136}$ and $Ag_{374}$ nanoclusters[22].

In recent research, chiral nanostructured materials have received a great deal of attention owing to their applications in chemistry, pharmacology, biology, and medicine[25–29]. Chirality refers to molecules that lack an internal plane of symmetry, and hence are non-superimposable with their mirror image. Recent studies have illustrated the potential applications offered by chiral metal nanoparticles in biomolecular recognition as powerful probes for macromolecules (e.g., proteins)[25]. Chiral noble metal nanostructures, in particular silver and gold, exhibit characteristic localized surface plasmon resonance, which leads to an intense optical activity[26, 27]. Chiral metal nanoparticles can also serve as a new route to enantioselective catalysis owing to their capability of enantioselective adsorption of chiral compounds, which is a key step toward achieving enantiospecific catalysis, separation, and sensing[28, 29]. However, preparation of ultrasmall-sized chiral nanoparticles with well-defined structures is still in the early stage of scientific advancement[29]. This is mainly owing to difficulties in the chemical synthesis of nanoparticles with intrinsic chiral morphology, or reliable assembly of achiral building blocks into chiral nanostructures, which have only been observed in the atomically precise gold nanoclusters[30–34].

Here, we present one-pot synthesis and crystal structure determination of a chiral silver nanocluster with 23 silver atoms capped by mixed organic ligands (i.e., phosphine and phenylethanethiolate), formulated as $Ag_{23}(PPh_3)_8(SC_2H_4Ph)_{18}$. The

structure is composed of two fcc unit cells with twist, hence, exhibiting helicity. This nanocluster also possesses an open shell electronic structure.

## Results

**Synthesis**. The $Ag_{23}(PPh_3)_8(SC_2H_4Ph)_{18}$ nanocluster was synthesized via a facile one-pot process. In brief, $AgNO_3$, phenylethanethiol, and $PPh_3$ were dissolved in a mixed solution of methanol and $CH_2Cl_2$. The Ag(I) species was reduced to Ag clusters using sodium borohydride. After a long "size-focusing" process at 4 °C, the initial, polydisperse silver clusters were eventually converged to monodisperse $Ag_{23}(PPh_3)_8(SC_2H_4Ph)_{18}$. Details of the synthesis are given in the Methods. The yield of the nanoclusters is ca. 10% (Ag atom basis). Single crystals of the clusters were obtained from a mixed solution of chloroform and methanol (3:1, v/v). The structure was solved by the X-ray crystallography (see details in the Supplementary Note 1 and Tables 1 and 2 ).

**Internal structure**. The $Ag_{23}(PPh_3)_8(SC_2H_4Ph)_{18}$ nanoclusters crystallize in the chiral monoclinic group $Cc$. The total structure is shown in Fig. 1. The Ag skeleton of the nanocluster is composed of 23 silver atoms ($Ag_{23}$), which can be seen as two joint cells in twisted fcc packing mode (Fig. 2b, c.f. Figure 2a for ideal fcc packing). The cells are twisted relative to each other by ca. 27° about the longitudinal axis of the cluster (Fig. 2b). This results in the deformation of the supercell's ends from a square to a rhombus-like shape (Supplementary Fig. 1). The rhombuses at two ends are not congruent; for example, their diagonals are slightly different in length (Supplementary Fig. 1). Two octahedra (Fig. 2c, highlighted in blue and red facets)—composed of 11 silver atoms of face centers of the two fcc units—are twisted by ca. 13° with respect to each other (Supplementary Fig. 2). The Ag–Ag interatomic distances within the nanocluster vary from 2.75 to 3.50 Å. The overall Ag framework of the nanocluster

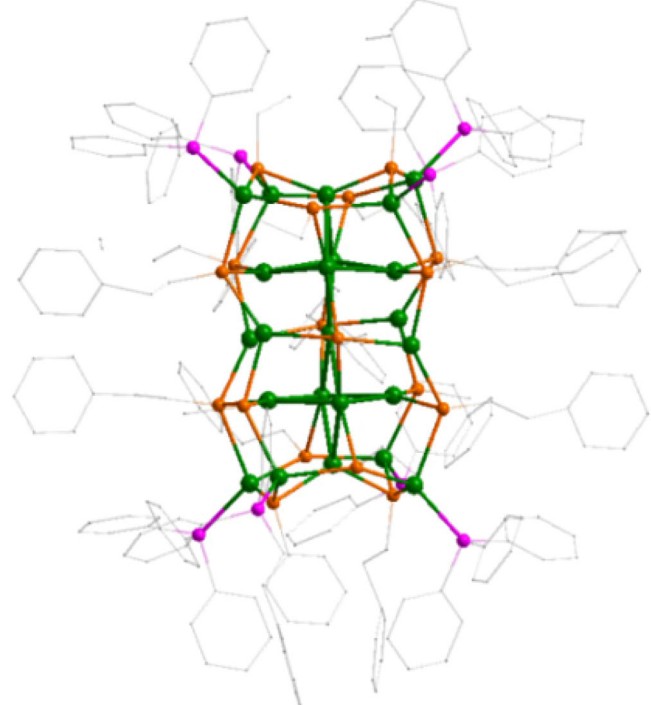

**Fig. 1** Total framework of the $Ag_{23}(PPh_3)_8(SC_2H_4Ph)_{18}$ nanocluster. Color codes: Ag, green; S, orange; P, purple; C, gray. Hydrogen atoms are omitted for clarity

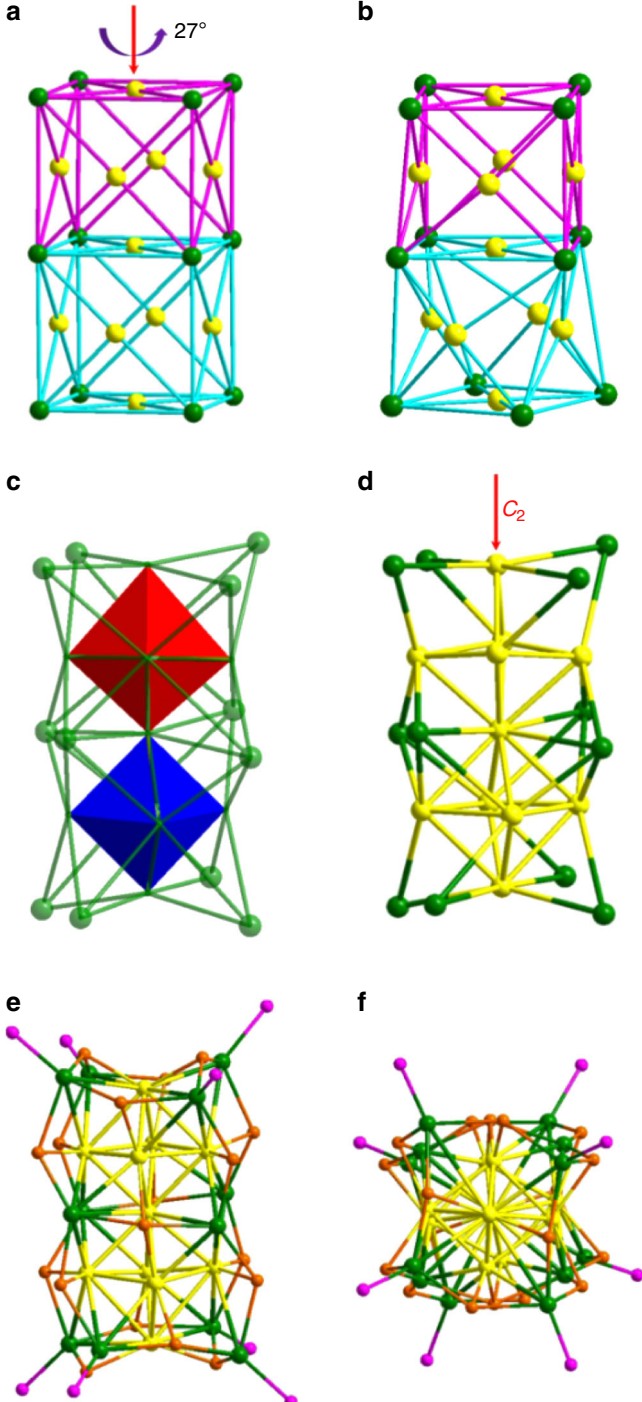

**Fig. 2** Anatomy of the twisted structure of $Ag_{23}(PPh_3)_8(SC_2H_4Ph)_{18}$ nanocluster. **a** Ideal structure of $1 \times 1 \times 2$ supercell of fcc packing mode. The face centers and vertices of the unit cells are shown in yellow and green, respectively. **b** Twisted supercell observed in the $Ag_{23}$ nanocluster, in which the two fcc unit cells are twisted by ca. $27°$ with respect to each other, resulting in a helical structure. **c** Two rhombic bipyramids within the nanocluster are highlighted by blue and red facets. **d** Total framework of $Ag_{23}$ with $C_2$ symmetry. **e** and **f** show the total framework of $Ag_{23}S_{18}P_8$ (top and side views). Color labels: S, orange; P, purple

adopts $C_2$ symmetry (Fig. 2d), therefore, $Ag_{23}$ has a chiral structure that originates from atomic arrangement of its metal core rather than configurations of the protecting ligands. Of note, the $C_2$ axis passes through the centers of the two fcc unit cells

(Fig. 2d). Eight phosphine ligands are bonded to Ag atoms at the vertices of the top and bottom of the nanocluster, with an average Ag–P distance of 2.45 Å (Fig. 2e, f, side and top views).

**Crystallographic arrangement**. The protecting thiolate ligands can be grouped into two main categories according to their binding patterns with the surface Ag atoms: (I) tri- and (II) tetrapodal types (Supplementary Fig. 3). The first category consists of three non-equivalent types of thiolate ligands on the surface of the nanocluster (subcategories IA, IB, and IC). As shown in Fig. 3a, a $-SR_{IA}$ ligand forms three bonds with two silver atoms (located at a unit cell's edge in the direction of the supercell's symmetry axis) and another Ag atom (at the face center of a facet containing this edge), with average Ag–S bond length being 2.54 Å. The two opposite facets are ligated by eight ligands of this type (Fig. 3a and Supplementary Fig. 4). It is worth noting that the face center Ag atom is shared by two $-SR_{IA}$ ligands facing each other.

The subcategory IB consists of four $-SR_{IB}$ ligands, each binding to two neighboring silver atoms at vertices at the end of the supercell with an average Ag–S bond length of 2.55 Å (Fig. 3b and Supplementary Fig. 5). In addition, each S atom forms a bond with a third Ag atom, located at the center of the side facet that contains the two Ag atoms (Fig. 3b).

There are four thiolate ligands belonging to subcategory IC. Two of them bind to the Ag atom located at the center of the cluster's top facet and two to Ag at the center of the bottom facet (Fig. 3c and Supplementary Fig. 6). Each of the four $-SR_{IC}$ ligands also forms bonds with two neighboring silver atoms at vertices of the top or bottom facet. The average Ag–S bond length of the subcategory IC is 2.57 Å.

Finally, there are two thiolate ligands, which are located at opposite sides of the supercell, that fall into category II (Fig. 3d and Supplementary Fig. 7). The S atoms of these $-SR_{II}$ ligands are tetra-coordinated to silver atoms—two situated at vertices and another two at centers of the joint unit cells. The average Ag–S bond length is 2.64 Å. The $-SR_{II}$ ligands are thus shared between two unit cells and can be viewed as hinges to hold the two fcc cells together. It is worth mentioning that the –R tails of the ligands are perpendicular to the facets they are attached to. The tetrapodal binding mode of thiolate ligands observed in the $Ag_{23}(PPh_3)_8(SC_2H_4Ph)_{18}$ cluster is a surprise. Such a coordination mode has not been reported previously. Of note, the central Ag atom (only one) at the center of the nanocluster does not coordinate to any ligand (i.e., the phosphine and thiolate).

**UV-vis and photoluminescence**. The optical adsorption spectrum of the nanocluster (in $CH_2Cl_2$, 0.1 mM) shows a strong peak at 515 nm and a weak tail band at 600–700 nm (Fig. 4a); the HOMO-LOMO gap is ca. 1.4 eV (Supplementary Fig. 8). Furthermore, the photoluminescence of the nanocluster was studied using excitation wavelengths of 350, 467, and 540 nm. The results show that the nanoclusters exhibit emission centered at ca. 800 nm regardless of the excitation wavelength (Fig. 4b and Supplementary Fig. 9).

**DFT simulation**. Time-dependent density functional theory (TDDFT) is applied to gain insight into the correlation between the nanocluster structure and its optical properties. To reduce the computational cost, we used $Ag_{23}(PH_3)_8(SCH_3)_{18}$ as a model for the nanocluster (see details in the Methods). Figure 4c shows the computed molecular orbitals (MOs). Of note, X-ray crystallographic analysis shows no counterion with the Ag nanocluster. This indicates that the number of free valance electrons of the nanocluster is odd ($23-18 = 5e$) and thus its electronic structure possesses an unpaired electron. To verify this, electron

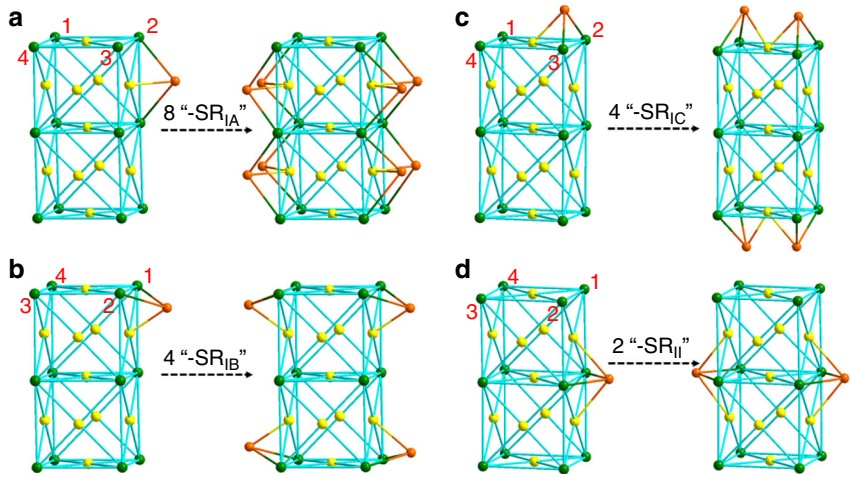

**Fig. 3** Binding patterns of thiolate ligands in subcategories. For the ease of visualization, the ideal structure (without twist) of Ag₂₃ is used, and four top vertices of the supercell are labeled by numbers

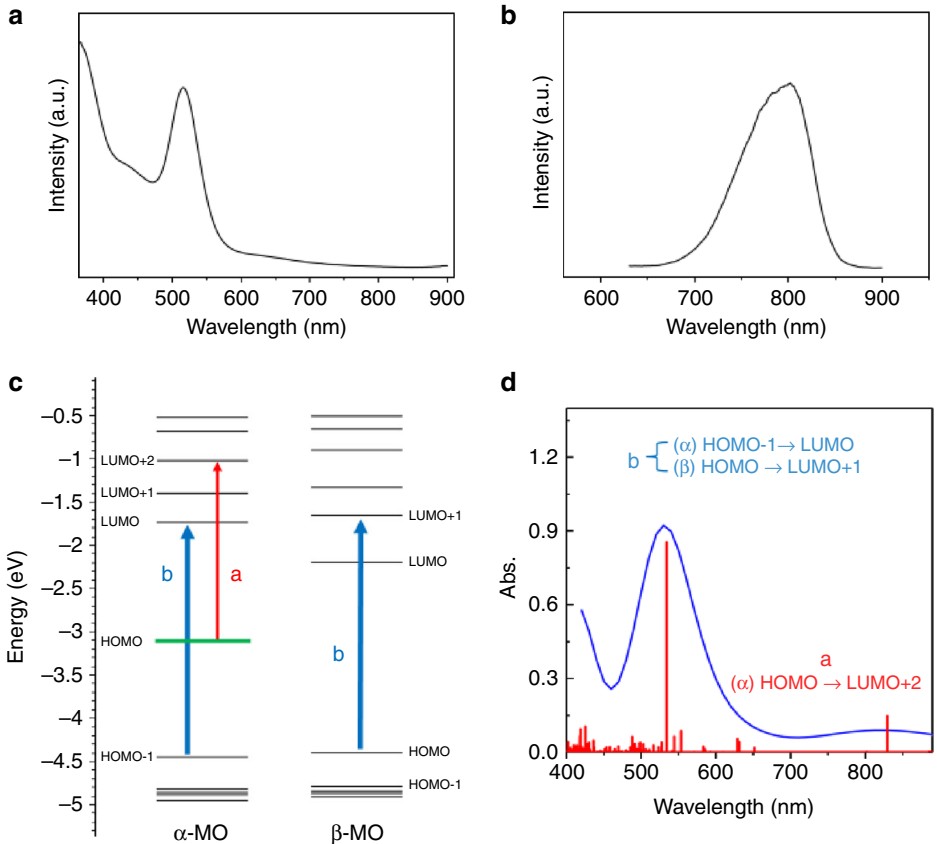

**Fig. 4** Optical properties and DFT simulation of the Ag₂₃ nanocluster. **a** UV-vis absorption and **b** emission ($\lambda_{ex}$ = 540 nm and slit width = 5 nm) spectra of Ag₂₃(PPh₃)₈(SC₂H₄Ph)₁₈ nanoclusters in a solution of CH₂Cl₂. **c** Energy alignment of the MOs and **d** theoretical absorption spectrum of Ag₂₃(PH₃)₈(SCH₃)₁₈

paramagnetic resonance (EPR) measurements[35–37] were performed. As shown in Supplementary Fig. 10, the EPR spectrum of the clusters (in CH₂Cl₂, at 110 K) exhibit one local maxima and one local minima, which is characteristic of a system with one unpaired electron ($s = \frac{1}{2}$), with $g = 1.959$ and 1.955. Therefore, open shell DFT calculations were performed that could reproduce the X-ray structure reasonably (Supplementary Table 1). Both $\alpha$ and $\beta$ orbitals are presented in Fig. 4c. The unpaired electron with the highest energy occupies HOMO of $\alpha$-MOs (shown in green).

Analysis of MOs composition indicates that the $\alpha$-HOMO is mainly localized on the waist silver atoms of the nanocluster—where two unit cells adjoin together (Supplementary Fig. 11). Analysis of atomic charges according to the CHelpG scheme[38] predicts that the central Ag atom carries a charge of −0.5 e, whereas all other silver atoms of the nanocluster are positively charged with ~0.2 e.

The optical absorption spectrum of the Ag nanocluster is calculated using TDDFT and fitted with Lorentzian functions

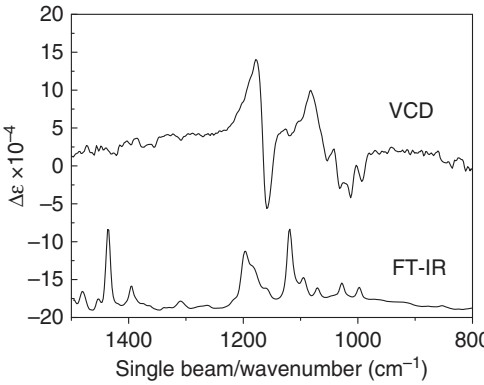

**Fig. 5** FT-IR and VCD spectra of the $Ag_{23}$ nanocluster. The IR peaks at 1179 and 1159 $cm^{-1}$ and 1082 and 1012 $cm^{-1}$ are assigned to the Ag–S–C and Ag–P–C stretching modes, respectively

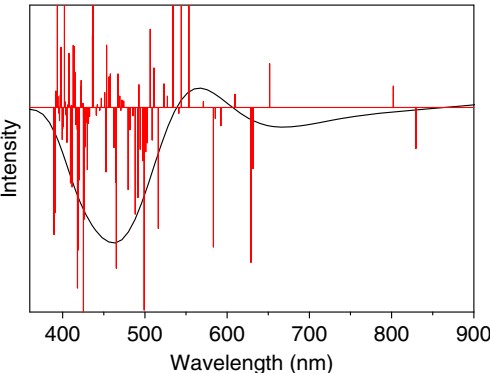

**Fig. 6** Simulated ECD spectrum of the $Ag_{23}$ nanocluster. The theoretical band at 527 nm is split into two bands at 465 and 570 nm via cotton effects

(Fig. 4d). The theoretical spectrum of the model is in good agreement with experimental results (Fig. 4a.d). There is an allowed electronic transition located at 829 nm, though its intensity is considerably lower than that of the main absorption peak at 527 nm. The 829 nm transition is attributed to electronic excitation of $\alpha$-HOMO → $\alpha$-LUMO + 2 (Fig. 4c, red arrow and labeled as 'a'). The main peak of the theoretical spectrum at ~527 nm is assigned to $\alpha$-HOMO-1 → $\alpha$-LUMO and $\beta$-HOMO → $\beta$-LUMO + 1 transitions (Fig. 4c, blue arrows and labeled as 'b'). These results indicate that the excitation of paired electrons is mainly responsible for the main absorption band. Population analysis of MOs shows that the $\alpha$-HOMO-1, $\alpha$-LUMO, $\beta$-HOMO, and $\beta$-LUMO + 1 are combinations of s and p atomic orbitals of Ag; therefore, the peak at ~527 nm essentially arises from $sp \rightarrow sp$ transitions (Supplementary Fig. 12).

**Vibrational and electronic circular dichroism**. Further, the chirality of the $Ag_{23}$ cluster is investigated by vibrational circular dichroism (VCD) technique. This spectroscopic technique provides a tool to identify the presence of chiral components through detection of differences in attenuation of left- and right-circularly polarized light[39]. VCD technique has already been employed to study the conformation of chiral thiolates on the surface of gold particles and clusters[40, 41]. Figure 5 shows the experimental IR and VCD spectra of $Ag_{23}$. A series of VCD peaks are observed in the range from 1200 to 1000 $cm^{-1}$, corresponding to the IR peaks. These IR peaks are assigned to the Ag–P–C and Ag–S–C stretching modes. Other IR regions show no VCD signals, implying that the observed VCD signals are from the twisted framework of the $Ag_{23}$ cluster. Although chiral HPLC isolation of $Ag_{23}$ enantiomers was not successful, we have performed theoretical simulations of the electronic circular dichroism (ECD) spectra of the $Ag_{23}$ cluster. The simulated ECD of $Ag_{23}$ clusters matches well with the corresponding UV-Vis absorption bands, and the Cotton effects split the theoretical band at 527 nm into two bands at 465 and 570 nm (Fig. 6). Taken together, these results clearly show the twist-induced chirality of the $Ag_{23}$ cluster.

## Discussion

In summary, we have devised a synthetic approach for $Ag_{23}(PPh_3)_8(SC_2H_4Ph)_{18}$ and solved its crystal structure. This chiral nanocluster comprises a helical fcc kernel protected by eight phosphine and 18 phenylethanethiolate ligands. Two coordination patterns for thiolate ligands are observed: tri- and tetra-podal types. By TDDFT calculations, we have further correlated the Ag nanocluster structure and its optical properties.

## Methods

**Synthesis of $Ag_{23}(SC_2H_4Ph)_{18}(PPh_3)_8$ nanoclusters**. $AgNO_3$ (22 mg, 0.13 mmol) was dissolved in 1 mL methanol solution. Then, 15 µL H-$SC_2H_4$Ph thiol (0.15 mmol) and 152 mg $PPh_3$ (0.58 mmol, dissolved in 5 mL $CH_2Cl_2$) were added. The mixed solution was stirred for 30 min at 600 rpm. A freshly prepared solution of $NaBH_4$ (8 mg, 0.21 mmol, dissolved in 1 mL cold water) was added drop by drop to reduce the Ag(I) species to Ag(0) nanoclusters. After reaction for 48 h, the organic phase was thoroughly washed with ethanol to remove excess thiol, phosphine, and salts. Then, pure $Ag_{23}(SC_2H_4Ph)_{18}(PPh_3)_8$ nanocluster was extracted with dichloromethane. Of note, the synthetic process was carried out at 10 °C under air. The yield of $Ag_{23}$ nanoclusters is ca. 10% (on Ag atom basis). Orange crystals of $Ag_{23}$ were obtained from the mixed solution of chloroform and methanol (3:1, v/v).

**Characterization**. The UV–vis absorption spectra were recorded on a Hewlett-Packard 8543 diode array spectrophotometer. X-ray diffraction data of $Ag_{23}(SC_2H_4Ph)_{18}(PPh_3)_8$ was collected on a Bruker X8 Prospector Ultra equipped with an Apex II CCD detector and an IµS micro-focus CuKα X-ray source ($\lambda = $ 1.54178 Å). Fluorescence spectra were recorded on a Fluorolog-3 spectrofluorometer (HORIBA Jobin Yvon). The EPR experiment was performed at 100 K, and $Ag_{23}(PPh_3)_8(SC_2H_4Ph)_{18}$ nanoclusters were dissolved in $CH_2Cl_2$ solutions (2 mM). The solution was introduced into a quartz tube before the EPR test. The VCD was performed on the ChiralIR-2 × (BioTools, Inc.). The $Ag_{23}$ clusters (1 mg) were mixed with 10 mg KBr, and then the mixture was pressed into a thin plate before the test.

**Computational details**. DFT optimization was performed using the B3PW91 hybrid density functional. The 6–31 G** basis set was employed for H, C, P, and S[42–45]. The LANL2DZ basis set was used for silver atoms[46]. TDDFT calculations of optical absorption spectra were performed and compared with experimental optical spectra. All calculations were carried out with the Gaussian09 package[45].

**Data availability**. The X-ray crystallographic coordinates for structures reported in this work (see Supplementary Tables 2 and 3 and Note 1) have been deposited at the Cambridge Crystallographic Data Centre (CCDC), under deposition number CCDC-1811890. These data can be obtained free of charge from The Cambridge Crystallographic Data Centre via www.ccdc.cam.ac.uk/data_request/cif.

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

## Acknowledgements

G.L. acknowledges financial support by the fund of the "Thousand Youth Talents Plan" and National Natural Science Foundation of China (No. 21601178). R.J. thanks the financial support from the Air Force Office of Scientific Research under AFOSR Award No. FA9550-15-1-9999 (FA9550-15-1-0154).

## Author contributions

G.L. and R.J. designed the study. C.L. and Z.L. synthesized the samples and carried out the testes. T.L. and C.Z. test the $Ag_{23}$ structure. H.A. and H.K. performed DFT simulations and analyses. G.L., H.A., and R.J. wrote the manuscript. All authors discussed the results and commented on the manuscript.

## Additional information

**Competing interests:** The authors declare no competing financial interests.

