## [Peer Review File · Nature Communications]

Reviewers' comments:

Reviewer #1 (Remarks to the Author):

Review of: "Chiral Ag₂₃ Nanocluster with Open Shell Electronic Structure and Helical Face-Centered Cubic Framework"

In this work, authors report the first Ag chiral thiolated cluster. Chirality has already been reported for the Au counterparts, but not on Ag. Chirality stems from a twist of an underlying fcc lattice framework. Additionally, authors find an unexpected S-Ag motif involving tetra-coordination (S bound to 4 Ag atoms). They also report optical absorption, emission and photocatalytic activity for the photo-oxidation of benzybenzamide to an imine.

The description of the structure is very well done, clear, and with detail.

I find the work publishable in Nature Communications; however, the authors should address some questions before further consideration.

1) The use of B3PW91 is a bit unusual in thiolated nanoclusters (at least for Ag). Is there a good justification to use it here? The accuracy in terms of the maximum absorption peak (527 nm for theory versus 515 nm for the experiment) seems to justify the method. However, it is not clear that HOMO-LUMO gaps are reasonably described. Is there any experimental determination of HOMO-LUMO gaps that they can use as a way of validation?

2) They do not mention anything about the computed structure. How well does it reproduce the x-ray structure? Please report comparison of important distances and the twist angles (exp. versus theory). For points 1 and 2, I realize that the letter format does not allow to include too much details in the main manuscript, but they can add this additional material in the supporting information.

3) The section describing catalysis is again a bit terse and not much discussion is included. It is clear that this cluster shows catalytic activity, but this doesn't tell me a lot about if it is better or worse (or much better or much worse) than, say, larger Ag nanoparticles. In other words, authors should discuss the catalytic activity relative to other systems.

4) Also, they should do a control experiment were only the TiO₂ support is assessed. In other words, how do they know TiO₂ is not acting as a catalyst as well? If they have done this control experiment, they should report it.

A minor point:

Use Helvetica or Arial for the axis labels and numbers in the Figure 4C.

Reviewer #2 (Remarks to the Author):

In this manuscript (NCOMMS-16-27857), the authors report that they have discovered a new silver nanoparticle (Ag₂₃) that is protected by both phosphine and thiol ligands. The cluster is chiral, displays a new binding motif, and has an open shell electronic configuration. These are all interesting results and nice work but it doesn't strike me as something important and surprising enough to belong in Nature Communications. New silver structures are now frequently discovered, and the discovery of a chiral cluster among them comes as no surprise. Chirality in gold nanoparticles is in fact quite

common. Further, gold clusters with an open electronic shell are also known, and silver there are clusters with even more surprising hydride ligands. The catalysis result is interesting but does not belong in this paper. Again, these are all very nice results, but is there a particular finding that makes this a very significant paper that demands immediate communication to the community? Will they strongly influence or change the direction of research in the field? I have difficulty answering these questions in the affirmative. The authors emphasize chirality and thereby make it the most important point of the paper, but again this is expected, therefore it is my opinion that this manuscript belongs in a more specialized journal.

I must also point out that referencing of other work is unacceptable for a manuscript of this nature. A more scholarly presentation would significantly improve the quality of the manuscript. Citation problems and other issues are listed below, along with other issues that should be addressed by the authors to improve their manuscript:

1 - The authors state: "significant efforts have been made to synthesize and characterize ultrasmall-sized Ag metalloid and Ag(I) nanoclusters.¹³⁻²⁴" but fail to cite the three most important early papers on silver clusters (Bakr, *ACIE* 2009; Cathcart, *Langmuir* 2009; Kumar, *JACS* 2010), which gave the field its start. This indicates that the authors are not as familiar with this field as they ought to be.

2 - The authors state: "As compared to the bulk counterpart, the nanoparticles possess higher surface-to-volume ratios and chemical potential. These characteristics lead to unique optical, electrical, and thermal properties..." This is not accurate. The confinement of electrons in these molecules and consequent electronic structure gives rise to their properties, not surface-to-volume ratios.

3 - The authors state: "Despite the great advances in the synthesis, the total structure determination of Ag nanoclusters still remains to be a major challenge. Thus far, only a few structures of Ag nanoclusters have been resolved by X-ray crystallography.¹⁶⁻²⁴" If this were stated a few years ago, it would have been accurate, however today the number of crystal structures of silver nanoclusters is too numerous to list without significant effort. Total structure determination of Ag nanoclusters is no longer what most would consider to be a "major challenge", although it is certainly of interest and worthy of significant efforts. The authors are missing quite a few structures from their list, which should include Ag₁₄, Ag₁₆, Ag₁₇, Ag₂₁, Ag₂₅, Ag₂₉, Ag₃₂, Ag₄₄, Ag₆₂, Ag₆₇, Ag₁₃₆, Ag₃₇₄, and their variants (I'm sure I missed a few, and I recognize that the authors have cited some of these).

4 - The authors state: "the plasmonic Ag nanoparticles are well known to adopt a face-centered cubic (fcc) structure." This is not well known. This demands a reference.

5 - The authors claim that chirality is rare in monolayer-protected metal clusters, however it is achirality that is rare. In a recent study by Garzon (*JPCA* 2015), 21 of 26 gold clusters (81%) were identified as chiral. Of the 21 chiral nanoparticles, 9 (or 43%) had chiral metal cores. The fact that a silver nanoparticle was found to be chiral is certainly significant, but it is not at all surprising. If this manuscript is about chirality, then the authors must familiarize themselves with this important work of Garzon.

6 - The synthetic experimental details are lacking, such that readers would not be able to reproduce the authors' work. For example, the authors state that a "long size-focusing process" was used to achieve the product, however it is not described in the manuscript. The authors also state that "Details of the synthesis are given in the Supporting Information." yet this information was entirely absent. This oversight needs to be corrected.

7 - With regard to the open electronic shell, this is certainly an interesting point of the work. The authors point to two observations, namely the EPR measurement and the failure to observe counterions in the crystal structure. The EPR measurement is no doubt important, however the fact that counterions were not detected in the xray data does not mean that they were not there. For example, if sodium ions were acting as counterions (derived from the sodium borohydride) then they would be exceptionally difficult to detect in a single-crystal x-ray diffraction measurement due to their small size and their relatively high mobility. It is possible to identify the alkali metal atoms by elemental analysis of the crystals. Using potassium instead of sodium would also be helpful for such experiments. Similar concerns for counter anions exist, but similar strategies could be used. Was mass spectrometry used to try to determine the charge on the nanoparticles? There are many well-developed techniques for analyzing charged and neutral species. These sorts of experiments would be an important verification of the EPR interpretation.

8 - In the conclusions, the statement that "The attainment of chiral Ag₂₃ nanoclusters without any chiral inducer (e.g. chiral ligand or solvent) provides a new avenue for future synthesis of chiral nanoclusters." is grossly overstated and inaccurate. This is simply the identification of one new species but there was no rationale provided that could be used to synthesize additional chiral silver nanoclusters. (Accidentally) discovering a synthetic method for synthesizing a chiral nanocluster does not alone constitute a "new avenue" for their synthesis.

9 - One final concern for now, and quite a significant one, is the fact that the authors have included catalysis results in a paper about the discovery and structure determination of a new silver nanoparticle. The catalysis results do not belong in this paper. They distract from the main results of the paper and in fact are relatively superficial, such that it is very difficult as a reviewer to scrutinize the results to the extent necessary to publish in a reputable journal. This catalysis study needs to be properly presented in its own paper, fully and completely, and refereed by experts in that particular field. It is one thing to present the structural and optical properties of the cluster, but it is quite another to begin to study chemical reactions. This makes the authors look desperate to add yet another result to a paper too weak to stand on its own, in hopes that it will make the report appear more substantial and impressive; this detracts from the work. The authors need to resist this way of thinking, remove the catalysis work, and fully develop and publish it separately.

Reviewer #3 (Remarks to the Author):

This paper reports the synthesis, structure determination and characterization of optical properties and photocatalysis of a silver nanocluster, formulated as Ag₂₃(PPh₃)₈(SC₂H₄Ph)₁₈. Unique findings of the paper are the construction of chiral Ag₂₃ structure using achiral ligands and open electronic structure with odd number of electrons (5e). The optical spectral features were explained using TDDFT theoretical calculations. Although photoluminescence and photocatalysis themselves are interesting and important, the readers will be more interested in new properties that may be originated from the unique structures of the clusters such as enantio-selective catalysis, optical resolution and CD activity, and/or electrochemistry. In this regard, I feel that the impact of this paper is limited. I regret to conclude that this paper is suitable for the publication in a more specialized journal rather than in Nature Communications.

1. Major concern for silver clusters is their stability against oxygen exposure and photo-irradiation. Please comment on (or show the evidence of) photostability in the presence and absence of oxygen. Were the clusters robust under the photocatalytic conditions?

2. According to Figure 3, four Ag atoms on the front and back facets are not bound by any ligands. If this understanding is correct, the authors should discuss the reason for this unique surface state.
3. Although it appears that 23 Ag atoms form a chiral (twisted) supershell structure, I think that the Ag atoms located at the corners and edges bound by thiolates (green balls in Figure 1D) should not be included as components of the core. This cluster should be viewed as face-sharing bi-octahedron capped by a variety of Ag(I)SR units.
4. What type of electronic excitations was induced by the photoabsorption at 455 nm? Please compare the TOF value with those of typical photocatalysts used for this reaction. The phrase "photocatalysis of benzylbenzamide" in the conclusion should read "photocatalytic oxidation of benzylbenzamide".
5. Please comment on the reason why mono-octahedral unit was not obtained, but only the dimer was obtained although it has open electronic structure?
6. It is interesting that the cluster has odd electron. This reminds me of the oxidation of anionic $\text{Au}_{25}(\text{SR})_{18}^-$ with 8e into a neutral state with 7e. Did the synthetic environment (in air or in inert gas) affect the charge state of the cluster products?

We thank the reviewers for their very helpful comments for improving our manuscript. Our point by point **response is in blue** and **revisions are in red** (also see the marked copy of manu. and SI).

Response to reviewers:

Reviewer #1:

I find the work publishable in Nature Communications; however, the authors should address some questions before further consideration.

1) The use of B3PW91 is a bit unusual in thiolated nanoclusters (at least for Ag). Is there a good justification to use it here? The accuracy in terms of the maximum absorption peak (527 nm for theory versus 515 nm for the experiment) seems to justify the method. However, it is not clear that HOMO-LUMO gaps are reasonably described. Is there any experimental determination of HOMO-LUMO gaps that they can use as a way of validation?

Response: The HOMO-LUMO gap of the Ag₂₃ is determined to be 1.4 eV using the spectrum on the energy scale (Figure S8 in the revised version of Supporting Information). This is in good agreement with the DFT results as HOMO-LUMO gap is calculated to be 1.3 eV. We also note the structural properties calculated by DFT are in good agreement with X-ray crystallographic analysis (Table S1 in the revised SI). These results indicate the use of B3PW91 functional could describe the experimental results quite well.

2) They do not mention anything about the computed structure. How well does it reproduce the x-ray structure? Please report comparison of important distances and the twist angles (exp. versus theory). For points 1 and 2, I realize that the letter format does not allow to include too much details in the main manuscript, but they can add this additional material in the supporting information.

Response: Thanks for the reviewer's suggestion. The experimental and theoretical structural properties (distances and the twist angles) of the Ag₂₃ cluster are compared and shown in the revised SI (Table S1). Results indicate that DFT could reproduce the X-ray structure reasonably.

Table S1. Comparison of the Ag-Ag distances and the twist angles of the Ag₂₃ cluster in experiment and theory.

distances and angles	experiment	theory
Ag-Ag in bipyramid	2.894 Å (2.754-3.295 Å)	2.933 Å (2.863-3.071 Å)
Ag-S (tetra-podal type)	2.635 Å (Ag-SC ₂ H ₄ Ph: 2.589-2.718 Å)	2.763 Å (Ag-SCH ₃ : 2.636-3.127 Å)
Ag-S (tri-podal type)	2.551 Å (Ag-SC ₂ H ₄ Ph: 2.394-2.711 Å)	2.641 Å (Ag-SCH ₃ : 2.526-2.783 Å)
Ag-P	2.462 Å (Ag-PPh ₃ : 2.412-2.506 Å)	2.620 Å (Ag-PH: 2.540-2.689 Å)
twist of cell (Ag ₁₄ unit)	27°	32°
twist of bipyramid	13°	17°

3) The section describing catalysis is again a bit terse and not much discussion is included. It is clear that this cluster shows catalytic activity, but this does not tell me a lot about if it is better or worse (or

much better or much worse) than, say, larger Ag nanoparticles. In other words, authors should discuss the catalytic activity relative to other systems.

Response: Thanks for raising this question. Our work reports for the first time the photo-catalytic conversion of benzylamine to imine by an Ag cluster. Our results show that the catalytic performance of the Ag₂₃ nanocluster is comparable or better than the previously reported systems (e.g., Au nanoparticles, Cu and Ru complexes, see Table S2).

Table S2. Catalytic activity for selective oxidation of benzylamine over various supported catalysts.

catalyst	T/°C	TOF ^a /h ⁻¹	reference
Au/TiO ₂	25	237	ACS Catal. 3 , 10, (2013).
Au/C	100	280	J. Catal. 264 , 138, (2009).
Au/CeO ₂	100	93.9	Chem. Commun. 50 , 292, (2014).
[Au ₂₅]NC/TiO ₂	30	878	ACS Catal. 7 , 3632, (2017).
Au@ZrO ₂	45	0.35	J. Am. Chem. Soc. 135 , 5793, (2013).
Ru ₂ (OAc) ₄ Cl	50	1.3	Syn. Lett. 11 , 1675, (2007).
CuCl	80	3.8	Bull. Chem. Soc. Jpn. 76 , 2399, (2003).
AgNC/P25	40	710	This work.

^aTOF of the reaction is calculated as (reacted mol of PhCH₂NH₂)/((mol of metal atom) × (reaction time)).

4) Also, they should do a control experiment where only the TiO₂ support is assessed. In other words, how do they know TiO₂ is not acting as a catalyst as well? If they have done this control experiment, they should report it.

Response: Thanks for reviewer's suggestion. The TiO₂ oxide also shows some catalytic activity for the reaction. TiO₂ gave 23% conversion in 90 min, but this activity is much lower than that of Ag₂₃/TiO₂ (62% conversion in 90 min), indicating the more important role of Ag₂₃. This info is now given in the revised manuscript.

A minor point:

Use Helvetica or Arial for the axis labels and numbers in the Figure 4C.

Response: We have corrected the font.

Reviewer #2:

1 - The authors state: “significant efforts have been made to synthesize and characterize ultrasmall-sized Ag metalloid and Ag(I) nanoclusters.13-24” but fail to cite the three most important early papers on silver clusters (Bakr, ACIE 2009; Cathcart, Langmuir 2009; Kumar, JACS 2010), which gave the field its start. This indicates that the authors are not as familiar with this field as they ought to be.

Response: We originally cited these authors’ more recent work (refs 17(Bigioni group) &20 (Bakr group), which we believe are more appropriate since the 2009/2010 papers are already well known and refs 17/20 are structure-related, whereas our manuscript is also structure-related. Speaking of early works, it is worth noting that Jin et al worked the first case of atomically precise Ag:thiolate clusters (*J Am Chem Soc* 2009), which was also left out because it’s already well known.

2 - The authors state: “As compared to the bulk counterpart, the nanoparticles possess higher surface-to-volume ratios and chemical potential. These characteristics lead to unique optical, electrical, and thermal properties...” This is not accurate. The confinement of electrons in these molecules and consequent electronic structure gives rise to their properties, not surface-to-volume ratios.

Response: Since we are talking about particles (as opposed to porous materials), the high surface to volume is equivalent to small size.

3 - The authors state: “Despite the great advances in the synthesis, the total structure determination of Ag nanoclusters still remains to be a major challenge. Thus far, only a few structures of Ag nanoclusters have been resolved by X-ray crystallography.16-24” If this were stated a few years ago, it would have been accurate, however today the number of crystal structures of silver nanoclusters is too numerous to list without significant effort. Total structure determination of Ag nanoclusters is no longer what most would consider to be a “major challenge”, although it is certainly of interest and worthy of significant efforts. The authors are missing quite a few structures from their list, which should include Ag14, Ag16, Ag17, Ag21, Ag25, Ag29, Ag32, Ag44, Ag62, Ag67, Ag136, Ag374, and their variants (I’m sure I missed a few, and I recognize that the authors have cited some of these).

Response: We believe it would better to balance the refs by citing a few from each research group — which we have done. Some of the listed species are Ag(I):L clusters, which are different from our current work. Again, it is not possible to cite all. But at least we have tried out best to balance various research groups.

4 - The authors state: “the plasmonic Ag nanoparticles are well known to adopt a facecentered cubic (fcc) structure.” This is not well known. This demands a reference.

Response: Both Ag and Au are well known to be fcc in the nano community.

5 - The authors claim that chirality is rare in monolayer-protected metal clusters, however it is achirality that is rare. In a recent study by Garzon (JPCC 2015), 21 of 26 gold clusters (81%) were identified as chiral. Of the 21 chiral nanoparticles, 9 (or 43%) had chiral metal cores. The fact that a silver nanoparticle was found to be chiral is certainly significant, but it is not at all surprising. If this manuscript is about chirality, then the authors must familiarize themselves with this important work of Garzon.

Response: We meant to say the silver case. Garzon’s paper is about gold. The different chemistries of Ag-S and Au-S surface bonding lead to different origins of chirality. For

example, in the current work we demonstrate the twist-induced chirality, which has not been observed in gold.

6 - The synthetic experimental details are lacking, such that readers would not be able to reproduce the authors' work. For example, the authors state that a "long size-focusing process" was used to achieve the product, however it is not described in the manuscript. The authors also state that "Details of the synthesis are given in the Supporting Information." yet this information was entirely absent. This oversight needs to be corrected.

Response: We have specified the time "48 hours". We tested the procedure by asking a new student to try out, who indeed reproduced the synthesis.

7 - With regard to the open electronic shell, this is certainly an interesting point of the work. The authors point to two observations, namely the EPR measurement and the failure to observe counterions in the crystal structure. The EPR measurement is no doubt important, however the fact that counterions were not detected in the x-ray data does not mean that they were not there. For example, if sodium ions were acting as counterions (derived from the sodium borohydride) then they would be exceptionally difficult to detect in a single-crystal x-ray diffraction measurement due to their small size and their relatively high mobility. It is possible to identify the alkali metal atoms by elemental analysis of the crystals. Using potassium instead of sodium would also be helpful for such experiments. Similar concerns for counter anions exist, but similar strategies could be used. Was mass spectrometry used to try to determine the charge on the nanoparticles? There are many well-developed techniques for analyzing charged and neutral species. These sorts of experiments would be an important verification of the EPR interpretation.

Response: Yes, the absence of counterion could be due to its disorder in the crystal. But we also did XPS and found no sodium ions. So our conclusion is still valid and we believe the current data is sufficient.

8 - In the conclusions, the statement that "The attainment of chiral Ag₂₃ nanoclusters without any chiral inducer (e.g. chiral ligand or solvent) provides a new avenue for future synthesis of chiral nanoclusters." is grossly overstated and inaccurate. This is simply the identification of one new species but there was no rationale provided that could be used to synthesize additional chiral silver nanoclusters. (Accidentally) discovering a synthetic method for synthesizing a chiral nanocluster does not alone constitute a "new avenue" for their synthesis.

Response: We respectively disagree! This single sentence is not a conclusion, but a **perspective** of future work (which is legitimate in all papers). This reviewer doesn't have to be upset!

9 - One final concern for now, and quite a significant one, is the fact that the authors have included catalysis results in a paper about the discovery and structure determination of a new silver nanoparticle. The catalysis results do not belong in this paper. They distract from the main results of the paper and in fact are relatively superficial, such that it is very difficult as a reviewer to scrutinize the results to the extent necessary to publish in a reputable journal. This catalysis study needs to be properly presented in its own paper, fully and completely, and refereed by experts in that particular field. It is one thing to present the structural and optical properties of the cluster, but it is quite another to begin to study chemical reactions. This makes the authors look desperate to add yet another result to a paper too weak to stand on its own, in hopes that it will make the report appear more substantial

and impressive; this detracts from the work. The authors need to resist this way of thinking, remove the catalysis work, and fully develop and publish it separately.

Response: We respectively disagree! *Nat Commun* is a multi-disciplinary journal and catalysis is one of the topics covered by the journal. We believe some audience of the journal would be interested in seeing photocatalysis. Having some applications such as photocatalysis is always a plus for a new material.

Reviewer #3:

This paper reports the synthesis, structure determination and characterization of optical properties and photocatalysis of a silver nanocluster, formulated as $\text{Ag}_{23}(\text{PPh}_3)_8(\text{SC}_2\text{H}_4\text{Ph})_{18}$. Unique findings of the paper are the construction of chiral Ag_{23} structure using achiral ligands and open electronic structure with odd number of electrons (5e). The optical spectral features were explained using TDDFT theoretical calculations. Although photoluminescence and photocatalysis themselves are interesting and important, the readers will be more interested in new properties that may be originated from the unique structures of the clusters such as enantio-selective catalysis, optical resolution and CD activity, and/or electrochemistry. In this regard, I feel that the impact of this paper is limited. I regret to conclude that this paper is suitable for the publication in a more specialized journal rather than in Nature Communications.

Response: Following the reviewer's suggestion we have carried out various measurements/simulations to explore the CD activity. Enantio-selective catalysis is very difficult and there has been no success yet with chiral clusters. We have measured vibrational circular dichroism (VCD) of Ag_{23} and found four peaks at 1178, 1159, 1082, and 1018 cm^{-1} (new Figure 5 added). The VCD signals arise from the twisted framework of the Ag_{23} cluster. Furthermore, electronic circular dichroism (ECD) of the Ag_{23} cluster is simulated using TDDFT (new Figure 6 added) since chiral HPLC separation of enantiomers was not successful. Taken together, our new results clearly show the twist-induced chirality of the Ag_{23} cluster.

(newly added) Figure 5. IR (lower profile) and VCD (upper one) spectra of the Ag_{23} cluster.

(newly added) Figure 6. Simulated ECD spectrum of the Ag_{23} cluster.

We have added the following discussion to the manuscript:

Further, the chirality of the Ag₂₃ cluster is investigated by vibrational circular dichroism (VCD) analysis. Figure 5 shows the experimental IR and VCD spectra of Ag₂₃. A series of VCD peaks are observed in the range from 1200 to 1000 cm⁻¹, corresponding to the IR peaks. These IR peaks are assigned to the Ag–P–C and Ag–S–C stretching modes. Other IR regions show no VCD signals, implying that the observed VCD signals are from the twisted framework of the Ag₂₃ cluster. While chiral HPLC isolation of Ag₂₃ enantiomers was not successful, we have performed theoretical simulations of the electronic circular dichroism (ECD) spectra of the Ag₂₃ cluster. The simulated ECD of Ag₂₃ clusters matches well with the corresponding UV-Vis absorption bands, and the Cotton effects split the theoretical band at 527 nm into two bands at 465 and 570 nm (Figure 6). Taken together, these results clearly show the twist-induced chirality of the Ag₂₃ cluster.

1. Major concern for silver clusters is their stability against oxygen exposure and photo-irradiation. Please comment on (or show the evidence of) photostability in the presence and absence of oxygen. Were the clusters robust under the photocatalytic conditions?

Response: Thanks for the reviewer’s suggestion. It is true that, compared to gold clusters, the Ag clusters are typically less robust. Nevertheless, the AgNC/TiO₂ catalyst did not lose the photocatalytic activity in three cycles, indicating its moderate stability.

2. According to Figure 3, four Ag atoms on the front and back facets are not bound by any ligands. If this understanding is correct, the authors should discuss the reason for this unique surface states.

Response: The viewing direction is turned by 90 degree in Figure 3B and 3D compared with Figure 3A and 3C. All the surface Ag atoms are bonded with the ligands (i.e., phosphine and thiolate).

3. Although it appears that 23 Ag atoms form a chiral (twisted) supershell structure, I think that the Ag atoms located at the corners and edges bound by thiolates (green balls in Figure 1D) should not be included as components of the core. This cluster should be viewed as face-sharing bi-octahedron capped by a variety of Ag(I)SR units.

Response: The definition of “core” is not unified in the literature. We agree that the reviewer’s picking of the bi-octahedron as the core is another way. We instead follow a traditional view of nanoparticles, i.e. *all metal atoms* being in the core and *organic ligands* as the shell, especially since in our case all the metal atoms contribute to chirality.

4. What type of electronic excitations was induced by the photoabsorption at 455 nm? Please compare the TOF value with those of typical photocatalysts used for this reaction. The phrase “photocatalysis of benzylbenzamide” in the conclusion should read “photocatalytic oxidation of benzylbenzamide”.

Response: Our LED lamp for photocatalysis emits at 455 nm. The electronic transition for 455 nm is higher than the gap absorption. We have compared the TOF value with other various supported catalyst (e.g., AuNP, Ru₂(OAc)₄Cl, and CuCl), Table S2. The TOF of AgNC/P25 is comparable or better than the previously reported systems (e.g., Au nanoparticles, Cu and Ru complexes, Table S2).

We have added Table S2 and also changed “photocatalysis of benzylbenzamide” to “photocatalytic oxidation of benzylamine” in the manuscript.

Newly added Table S2. Catalytic activity for selective oxidation of benzylamine over various supported catalysts.

catalyst	T/°C	TOF ^a /h ⁻¹	reference
Au/TiO ₂	25	237	ACS Catal. 3 , 10, (2013).
Au/C	100	280	J. Catal. 264 , 138, (2009).
Au/CeO ₂	100	93.9	Chem. Commun. 50 , 292, (2014).
[Au ₂₅]NC/TiO ₂	30	878	ACS Catal. 7 , 3632, (2017).
Au@ZrO ₂	45	0.35	J. Am. Chem. Soc. 135 , 5793, (2013).
Ru ₂ (OAc) ₄ Cl	50	1.3	Syn. Lett. 11 , 1675, (2007).
CuCl	80	3.8	Bull. Chem. Soc. Jpn. 76 , 2399, (2003).
AgNC/P25	40	710	This work.

^aTOF of the reaction is calculated as (reacted mol of PhCH₂NH₂)/((mol of metal atom) × (reaction time)).

5. Please comment on the reason why mono-octahedral unit was not obtained, but only the dimer was obtained although it has open electronic structure?

Response: This is an interesting question but hard to address. Generally, size control is owing to kinetic control of the early-stage product in the synthesis and the final stable size survived is however largely determined by thermodynamics.

6. It is interesting that the cluster has odd electron. This reminds me of the oxidation of anionic Au₂₅(SR)₁₈⁻ with 8e into a neutral state with 7e. Did the synthetic environment (in air or in inert gas) affect the charge state of the cluster products?

Response: The synthetic process was carried out in **air** at a relatively low temperature (ca. 10 °C). The atmosphere (e.g., in air or N₂) did not affect the charge state of the cluster product. And, Ag₂₃ cluster couldn't be further reduced by NaBH₄ to Ag₂₃⁻ with 6e closed shell.

Revision: we have added a sentence (page 8) "Of note, the synthetic process was carried out under air at 10 °C."

Reviewers' comments:

Reviewer #1 (Remarks to the Author):

Author's responses to my inquires and suggestions are acceptable.

Reviewer #3 (Remarks to the Author):

I found that the authors properly addressed most of the questions and concerns raised by the reviewers. However, I could not agree with the authors concerning the following points.

1. In the revised version, the authors reported that the Au₂₃/TiO₂ catalysts showed comparable or better performance in photo-oxidation of benzylbenzamide than other supported catalysts and that the catalysts could be reused at least three times. However, I think that the results on photocatalysis should not be included in this paper since the high catalytic performance is not based on the novel features of the Ag₂₃ clusters (chirality and open electronic structure). The report of superior catalysis without any explanation of the reasons will not merit the readers.

2. Although this paper reports the first example of chiral Ag clusters protected by achiral ligands, the construction of chiral nanostructures using achiral building units is well known. In addition, this paper does not provide any general method to construct chiral structures from achiral components.

Therefore, the following phrase on the future prospect sounds overstated.

"The attainment of chiral Ag₂₃ nanoclusters without any chiral inducer provides a potential new avenue for controlled synthesis of new chiral nanoclusters in future work"

Minor point:

1. I would like to suggest the authors cite the following recent paper on single crystal XRD study on enantiopure chiral Au clusters in the second paragraph in page 3.

S. Takano, T. Tsukuda, Amplification of the optical activity of gold clusters by the proximity of BINAP. J. Phys. Chem. Lett. 7, 4509 (2016).

2. Since vibrational circular dichroism (VCD) has not been widely used as a characterization method of nanoclusters, please provide a general description on VCD for heterogeneous readers who are not familiar with the method. I do not understand how the racemic mixture of enantiomers exhibits VCD signals. Peaks in the IR spectrum (Figure 5) should be assigned explicitly.

Reviewers' comments:

Reviewer #1 (Remarks to the Author):

Author's responses to my inquires and suggestions are acceptable.

Response: We thank the reviewer for the comment.

Reviewer #3 (Remarks to the Author):

I found that the authors properly addressed most of the questions and concerns raised by the reviewers. However, I could not agree with the authors concerning the following points.

1. In the revised version, the authors reported that the Au₂₃/TiO₂ catalysts showed comparable or better performance in photo-oxidation of benzylbenzamide than other supported catalysts and that the catalysts could be reused at least three times. However, I think that the results on photocatalysis should not be included in this paper since the high catalytic performance is not based on the novel features of the Ag₂₃ clusters (chirality and open electronic structure). The report of superior catalysis without any explanation of the reasons will not merit the readers.

Response: We have removed the photocatalysis section.

2. Although this paper reports the first example of chiral Ag clusters protected by achiral ligands, the construction of chiral nanostructures using achiral building units is well known. In addition, this paper does not provide any general method to construct chiral structures from achiral components. Therefore, the following phrase on the future prospect sounds overstated. "The attainment of chiral Ag₂₃ nanoclusters without any chiral inducer provides a potential new avenue for controlled synthesis of new chiral nanoclusters in future work"

Response: We have removed the phrase.

Minor point:

1. I would like to suggest the authors cite the following recent paper on single crystal XRD study on enantiopure chiral Au clusters in the second paragraph in page 3.

S. Takano, T. Tsukuda, Amplification of the optical activity of gold clusters by the proximity of BINAP. J. Phys. Chem. Lett. 7, 4509 (2016).

Response: The paper has been cited as ref. 34 in revised version of the manuscript.

2. Since vibrational circular dichroism (VCD) has not been widely used as a characterization method of nanoclusters, please provide a general description on VCD for heterogeneous readers who are not familiar with the method. I do not understand how the racemic mixture of enantiomers exhibits VCD signals. Peaks in the IR spectrum (Figure 5) should be assigned explicitly.

Response: We have added a general description of vibrational circular dichroism. Additionally, IR peaks are assigned in the revised version of the manuscript. We have added the following discussion to the manuscript:

“This spectroscopic technique provides a tool to identify the presence of chiral components through detection of differences in attenuation of left- and right-circularly polarized light.³⁹ The VCD technique has already been employed to study the conformation of chiral thiolates on the surface of gold particles and clusters.^{40,41}”